# Discrimination between Breast Cancer Cells and White Blood Cells by Non-Invasive Measurements: Implications for a Novel In Vitro-Based Circulating Tumor Cell Model Using Digital Holographic Cytometry

**Zahra El-Schich** [1,2,*], **Birgit Janicke** [3], **Kersti Alm** [3], **Nishtman Dizeyi** [4], **Jenny L. Persson** [1,2,5] **and Anette Gjörloff Wingren** [1,2]

1 Department of Biomedical Sciences, Faculty of Health and Society, Malmö University, 205 06 Malmö, Sweden; jenny.persson@umu.se (J.L.P.); anette.gjorloff-wingren@mau.se (A.G.W.)
2 Biofilms-Research Center for Biointerfaces, Malmö University, 205 06 Malmö, Sweden
3 Phase Holographic Imaging AB, 223 63 Lund, Sweden; birgit.janicke@phiab.se (B.J.); kersti.alm@phiab.se (K.A.)
4 Department of Translational Medicine, Lund University, 205 06 Malmö, Sweden; nishtman.dizeyi@cernelle.se
5 Department of Molecular Biology, Umeå University, 901 87 Umeå, Sweden
* Correspondence: zahra.el-schich@mau.se

**Abstract:** Breast cancer is the second most common cancer worldwide. Metastasis is the main reason for death in breast cancer, and today, there is a lack of methods to detect and isolate circulating tumor cells (CTCs), mainly due to their heterogeneity and rarity. There are some systems that are designed to detect rare epithelial cancer cells in whole blood based on the most common marker used today, the epithelial cell adhesion molecule (EpCAM). It has been shown that aggressive breast cancer metastases are of non-epithelial origin and are therefore not always detected using EpCAM as a marker. In the present study, we used an in vitro-based circulating tumor cell model comprising a collection of six breast cancer cell lines and white blood cell lines. We used digital holographic cytometry (DHC) to characterize and distinguish between the different cell types by area, volume and thickness. Here, we present significant differences in cell size-related parameters observed when comparing white blood cells and breast cancer cells by using DHC. In conclusion, DHC can be a powerful diagnostic tool for the characterization of CTCs in the blood.

**Keywords:** breast cancer; cell area; cell thickness; cell volume; circulating tumor cell; CD45; digital holographic cytometry; EpCAM

## 1. Introduction

Breast cancer is the leading cause of cancer deaths among women worldwide [1]. Breast cancer metastasis accounts for the majority of deaths from breast cancer. The detection of metastases at the earliest stage is important for the management and estimation of breast cancer progression [2]. Breast cancer treatments today are based on the absence or presence of the hormone estrogen receptor (ER), the progesterone receptor (PR) and the expression of human epidermal growth factor receptor 2 (HER2) [3,4]. A tumor with the absence of all of these three receptors, also called triple negative breast cancer, is an aggressive form of breast cancer with a high risk of relapse and metastasis, and is associated with a poor clinical outcome [5–7].

Epithelial cells may undergo epithelial-to-mesenchymal transition (EMT), allowing the cells to gain new abilities to cross the extracellular matrix, migrate and become circulating cells [8,9]. The presence of circulating tumor cells (CTC) in patients with metastatic breast cancer is indicative of poor prognosis [10]. The survival rate of women with breast cancer is highly dependent on the absence of metastatic cells and the tumor grade [2].

The most common marker for CTCs that is used today is the epithelial cell adhesion molecule (EpCAM), but for some malignancies such as aggressive breast cancer metastases that are often of non-epithelial origin or have transitioned through EMT, the EpCAM detection method is not useful [11]. This highlights the need for more reliable CTC markers. The adhesion receptor CD44 and EpCAM function together in preparing the pre-metastatic niche [12], while CD45 is a leukocyte marker that is exclusively expressed on all nucleated cells of the hematopoietic system. The absence of CD45 on epithelial cells emphasizes the fact that it is an appropriate cell surface marker for distinguishing between CTCs and hematopoietic cells [13].

CTCs are known to have morphological properties that distinguish them from other circulating cells, such as a larger size than leukocytes and different nuclear morphology [14]. CTCs have been isolated by size, using different methods such as density gradient centrifugation and micropore filtration [15–19]. A more cell-friendly and speedy method of investigating morphological differences would facilitate CTC research. Digital holographic cytometry (DHC) is a powerful tool for label-free cell observations and the evaluation of cell morphological and dynamical parameters in vitro [20–26]. Studies using DHC include different cell types, from protozoa, bacteria and plant cells to mammalian cells such as nerve cells, stem cells and tumor cells [23]. DHC has also become popular in the diagnostic field such as for the screening of malaria-infected red blood cells and cervical cancer [21,27–29].

The aim of the present study was to use DHC to investigate the morphological differences between a collection of breast cancer cell lines and white blood cell (WBC) cancer cell lines. The flow cytometry analysis of the cell markers EpCAM and CD45 was used to characterize the cell lines. We show that cell types lacking CD45 expression have significantly larger cell area, volume and thickness than $CD45^+$ cells without EpCAM expression. In conclusion, DHC is a powerful technique for discriminating cancer cells from WBCs by performing non-invasive measurements of cell area, volume and thickness.

## 2. Materials and Methods

### 2.1. Cell Culture

The human breast cancer cell lines Hs-578T, MDA-MB-231, MDA-MB-468, T-47D, Cama-1 and MCF7 and WBC cell lines human leukemic lymphocyte Jurkat and human leukemic monocyte THP-1 were obtained from the American Type Culture Collection (ATCC/LGC Standards, Teddington, UK). Hs-578T cells were cultured in Dulbecco's Modified Eagle Medium (DMEM, Invitrogen, San Diego, CA, USA) supplemented with 10% fetal bovine serum (FBS, Invitrogen) 1% penicillin–streptomycin and 10 µg/mL of insulin (Sigma-Aldrich, St. Louise, MO, USA). MDA-MB-231 and MDA-MB-468 cells were cultured in DMEM supplemented with 10% FBS. T-47D, MCF7, Jurkat and THP-1 cells were cultured in Roswell Park Memorial Institute (RPMI-1640, Invitrogen) medium supplemented with 10% FBS and 50 µg/mL of gentamycin (Invitrogen). Cama-1 cells were cultured in RPMI-1640 medium supplemented with 10% FBS, 1% pyruvate sodium (Invitrogen) and 1% penicillin–streptomycin (Invitrogen). The cell lines were incubated at 37 °C with 5% $CO_2$ and 95% humidity.

### 2.2. Flow Cytometry Analysis

EpCAM staining: $1 \times 10^6$ cells/sample were stained with anti-EpCAM-PE (Miltenyi Biotec GmBH, Bergisch Gladbach, Germany) or left unstained as a control. The cells were washed twice with 2 mL of phosphate buffered saline (PBS, Invitrogen). A volume of 100 µL of anti-EpCAM-PE (10 ng/mL in PBS) was added to the cells, and 100 µL of PBS was used as a negative control. The cells were incubated

with EpCAM-PE for 30 min at 4 °C in the dark. After incubation, the cells were washed three times with 2 mL of PBS and analyzed using flow cytometry (BD Accuri C6, NJ, USA).

CD45 staining: $1 \times 10^6$ cells/sample were stained with human anti-CD45-FITC antibody (BD Biosciences, San Jose, CA, USA) or left unstained as a control. The cells were washed twice with 2 mL of PBS, and thereafter, 100 μL of anti-CD45-FITC (10 μg/mL in PBS) was added. 100 μL of PBS was used as a negative control. The cells were then incubated for 30 min at 4 °C in the dark. After incubation, the cells were washed twice with 2 mL of PBS and analyzed using flow cytometry (BD Accuri C6).

### 2.3. DHC and Computer Software

Cell morphology was detected and analyzed using DHC with the HoloMonitor™ M4 (Phase Holographic Imaging AB, PHIAB, Lund, Sweden). For each analysis, $1 \times 10^6$ cells in suspension were washed twice with PBS and then $1 \times 10^4$ cells in 10 μL were added to a Countess™ cell counting chamber slide (ThermoFisher Scientific, Waltham, MA, USA). The chamber slide was placed on a HoloMonitor™ M4 inside a cell incubator to ensure stable conditions for the cells, 37 °C with 5% $CO_2$ and 95% humidity for 10 min, during the analysis. For each cell line, more than 10 images at different positions were acquired automatically all over the chamber (Figure 1). The image analysis was performed with the proprietary AppSuite software (Phase Holographic Imaging AB). The HoloMonitor™ M4 was equipped with a 635 nm diode laser, which illuminated the cells at 0.2 mW/cm² to prevent phototoxic effects on the cells.

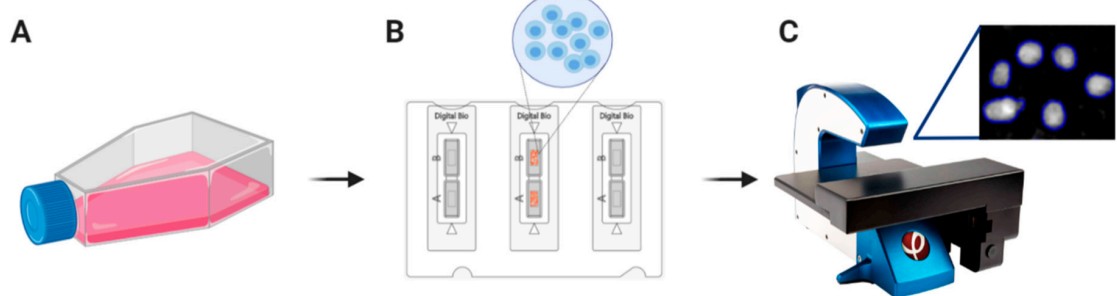

**Figure 1.** A schematic flow of the experimental system that consisted of the cultured cells (**A**), the detached cells in the cell counting chamber slides (**B**) and the HoloMonitor™ M4 analysis set-up (**C**).

### 2.4. Statistical Analysis

The cell morphology was analyzed by calculating the mean of at least 10 images of the cell mean volume, mean area and mean thickness with a cell number average of 242 ± 58 cells in each image. Experiments were repeated on separate days, at least 3 times. The analyzed data of at least 10 images per sample for DHC are presented as mean ± standard deviation (STDEV). Statistical significance was determined by two-tailed unpaired Student's t-tests, with P values ≤0.05 considered significant.

## 3. Results

### 3.1. There Is Low or No Expression of EpCAM in Triple Negative Breast Cancer Cell Lines

EpCAM expression is limited to normal and malignant epithelial cells. EpCAM has been used as a diagnostic marker for the detection of carcinoma cells in mesenchymal organs such as the blood, bone marrow or lymph nodes [30]. In this study, all the cell lines were first classified by analyzing EpCAM and CD45 expression. The three triple negative breast cancer cell lines;Hs-578T, MDA-MB-231 and MDA-MB-468, expressed no or little EpCAM protein, while the three non-metastatic cell lines, T-47D, Cama-1 and MCF7, expressed high levels of EpCAM (Table 1). THP-1 and Jurkat showed no

EpCAM expression but expressed high levels of CD45. All the breast cancer cell lines were negative for CD45 (Table 1).

**Table 1.** EpCAM and CD45 expression on different cancer cell lines were analyzed with flow cytometry. The results are from one representative experiment out of three, and the results are presented as % positive cells.

|  | Jurkat | THP-1 | Hs-578T | MDA-MB-231 | MDA-MB-468 | T-47D | Cama-1 | MCF7 |
|---|---|---|---|---|---|---|---|---|
| *EpCAM %* | 0.2 | 0.1 | 0.1 | 0.3 | 11.7 | 48.6 | 79.9 | 91.4 |
| CD45 % | 100 | 100 | 0.1 | 0.4 | 0.1 | 1.2 | 1.2 | 0.9 |

*3.2. WBC Lines and Breast Cancer Cell Lines Presented in 2D Holograms with DHC*

The two WBC lines THP-1 and Jurkat, and the six different breast cancer cell lines were analyzed for cell mean volume, mean area and mean thickness with DHC (Figures 2 and 3). As shown here, the WBC cell lines THP-1 and Jurkat were consistently smaller than the six breast cancer cell lines.

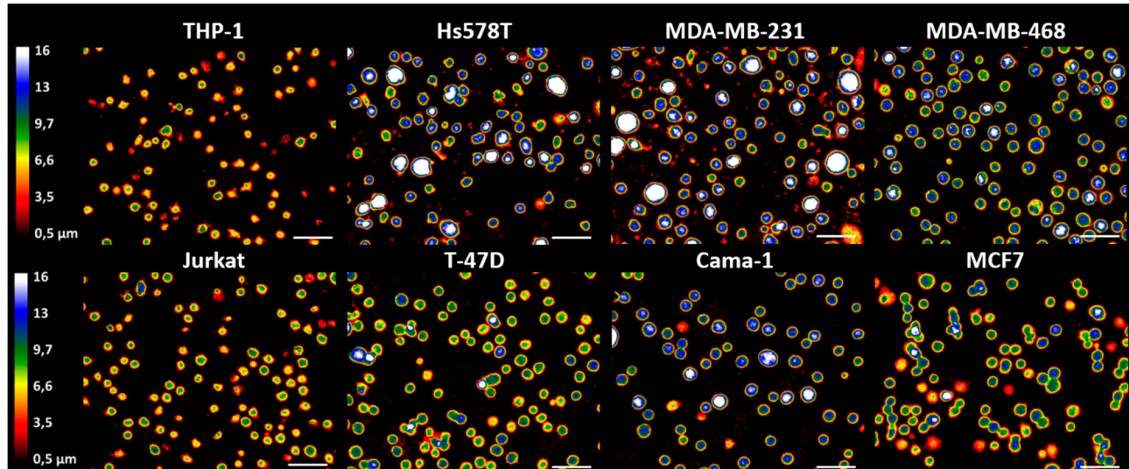

**Figure 2.** DHC holographic phase image in 2D. Two white blood cell lines (THP-1 and Jurkat) and six different breast cancer cell lines were analyzed with DHC. The color scale bars represent the thickness of the cells in μm, and the scale bars represent 10 μm.

*3.3. The Morphology Parameters Analyzed with DHC Clearly Discriminate between Breast Cancer Cells and WBC Lines*

As measured by DHC, the WBC lines Jurkat and THP-1 cells have smaller cell areas, cell volumes and cell thicknesses than the six breast cancer cell lines. Indeed, there were significant differences in the cell area, cell volume and cell thickness when comparing the WBC lines and breast cancer cell lines, as calculated by two-tailed unpaired Student's t-tests (Figure 3a–c).

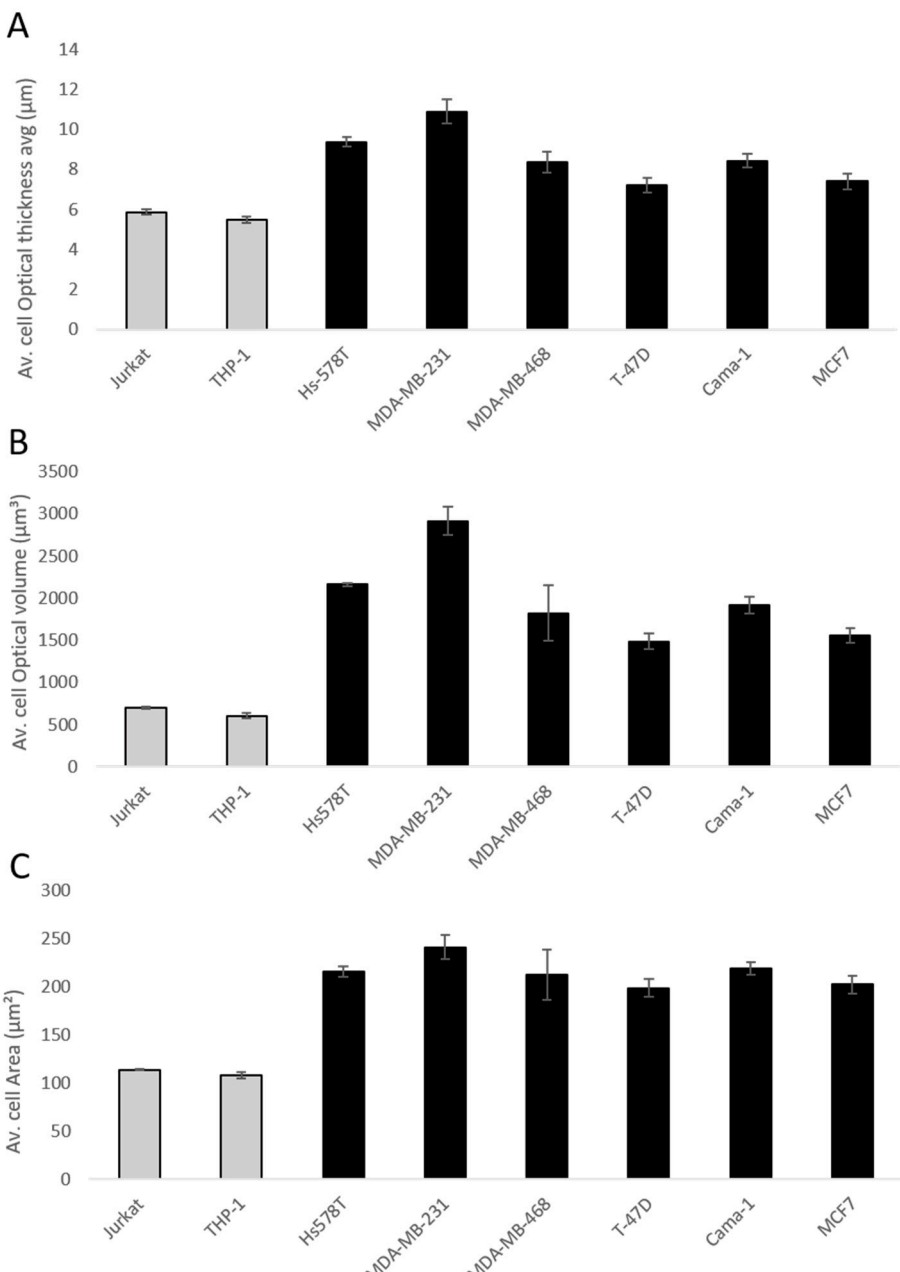

**Figure 3.** Diagrams presenting the average cell area (**A**), average optical volume (**B**) and average cell thickness (**C**) for Jurkat, THP-1 and six breast cancer cell lines. The calculations were based on at least 10 images/sample. All results comparing THP-1 and Jurkat with the breast cancer cells were significant with p < 0.005.

## 4. Discussion

In 2004, the CellSearch system was approved for the detection of CTCs in humans by the Food and Drug Administration (FDA). This system is based on magnetic cell sorting, the expression of EpCAM and cytokeratin (CK) [31]. The system is reliable and has radically improved the detection of CTCs. However, one drawback is the inability to detect CTCs with downregulated EpCAM expression. The level of EpCAM varies in different tumor types and is known to be downregulated to increase invasiveness and metastatic potential [11,32]. The CellSearch system may miss EpCAM-negative tumor cells due to the loss of EpCAM expression. Several sorting methods exist based on other cell characteristics, such as physical properties (size, elasticity and surface charge). CTCs are usually larger

in size, in contrast to leukocytes, and therefore, many CTCs isolation methods are mainly based on size differences [14,33,34]. A microcavity array (MCA) system based on a polycarbonate filter for the isolation by size of epithelial tumor cells has been developed [35,36]. The MCA system showed a significantly higher degree of CTCs isolation in advanced lung cancer patients than the CellSearch system [35]. Ribeiro-Samy et al. developed a microfluidic chip (CROSS) for the label-free isolation of CTCs, which showed higher sensitivity than CellSearch when isolating CTCs from metastatic colorectal cancer patients [36].

As the morphological parameters including the area, thickness and volume of WBCs are distinctly different from epithelial cancer cell morphology, a CTC detection system based on cell morphology would detect even EpCAM-negative cancer cells. DHC is label-free and provides convenient quantitative imaging analyses of cell morphology for both single cells and cell populations. DHC applications have previously been used to discriminate between the maturity levels of red blood cells, perform cell death studies and analyze drug responses [23,37–39]. Cell optical area, thickness and volume are useful parameters, particularly for discrimination between different cell types [39–44].

Recent developments in microfluidic methods have allowed a number of new investigations related to CTCs; however, the low frequency of CTCs in the circulation remains a challenge for the application of CTC-based technologies in the clinical setting [45]. DHC could, in combination with machine-learning algorithms, distinguish between normal skin cells and melanoma cells, and between primary and metastatic melanoma cell lines with high significance [46]. Machine-learning algorithms and computational tools are now provided methods for optimizing DHC for kinetic single adherent cell classification [47]. Singh et al. optimized high-throughput holographic screening for detecting tumor cells in blood through a microchannel system and making it a promising tool for the label-free analysis of liquid biopsy samples [48]. Khan et al. merged two technologies to perform the continuous high-resolution fluorescence imaging of cellular suspensions in a deep microfluidics chamber [49]. Here, a microfluidics chamber could be one option for continuing this project and extending the analysis to a large number of WBCs.

In this study, we have shown that epithelial breast cancer cells can be discriminated from WBCs by measuring the cellular thickness, volume and area using DHC. The investigated parameters showed significant differences between epithelial cells and blood cells. Future applications will include peripheral blood cells and extended collections of epithelial cancer cells.

## 5. Conclusions

Non-invasive DHC is a powerful technique to use for the discrimination of epithelial cancer cells and WBCs. Upon determining the cell area, volume and thickness, the investigated cell types with CD45 expression, the WBCs, were shown to have significantly smaller cell areas, volumes and thicknesses than CD45$^-$ epithelial cells either with or without EpCAM expression. We conclude that DHC is a convenient technique with many possibilities for analyzing either suspension or adherent cells, here used in a model for the in vitro analysis of CTCs.

**Author Contributions:** Methodology, software and data curation, Z.E.-S., K.A. and A.G.W.; writing—original draft preparation, Z.E.-S. and A.G.W.; writing—review and editing, K.A., B.J., N.D. and J.L.P.; supervision, A.G.W. All authors have read and agreed to the published version of the manuscript.

**Funding:** This research was funded by the Swedish Knowledge Foundation, the European Union´s Horizon 2020 research and innovation program under the Marie Sklodowska-Curie grant agreement grant number 721297, Biofilms Research Center for Biointerfaces and Malmö University.

**Acknowledgments:** We thank Bo Holmqvist and Anders Brinte at ImaGene-iT, Lund, Sweden; Karin von Wachenfeldt at Truly Translational AB, Sweden; and Börje Sellergren at the Department of Biomedical Sciences, Faculty of Health and Society, Biofilms-Research Center for Biointerfaces, Malmö University, Malmö, Sweden.

**Conflicts of Interest:** The authors declare no conflict of interest. Kersti Alm and Birgit Janicke are employed by Phase Holographic Imaging, who manufacture HoloMonitor M4.

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
