# Peer review of "Discrimination between Breast Cancer Cells and White Blood Cells by Non-Invasive Measurements: Implications for a Novel In Vitro-Based Circulating Tumor Cell Model Using Digital Holographic Cytometry"

_applsci, doi:10.3390/app10144854_

Round 1
Reviewer 1 Report
It is a well written communication piece on using DHC to discriminate between cells by non-invasive methods. But I believe more comparative study referring other scientific articles and including more necessary information would make this paper publishable in applied sciences journal. Please find below questions which should be addressed in the paper.
- Author should include more references where other research articles used DHC as a mean to classify cells. Such as: Hejna et al, Scientific Reports volume 7, Article number: 11943 (2017). And also author needs to quantitatively specify how the current method is better than other existing studies.
- It would make the paper better if author can extend the applicability of this DHC study in microfluidic flow system where cells can move continuously in deep chamber, instead of being stagnant on a slide. Handing large sample by continuous flow is important for rare cell study. A very related article in Analytical Chemistry was published where WBC were flowed on a chip for imaging cytometry applications (Khan et al, Anal. Chem. 2018, 90, 13, 7862–7870). Author may want to refer this paper and discuss whether DHC study system can be extended in microfluidic system as well.
- A complete schematic / flow chart of the experimental system is necessary here to give a clear picture to the readers.
- What was the sample number over which cell thickness, area, and volume was measured? This need to be included.
- Is there any other morphological parameters obtainable from DHC to compare between two types of cell? I was wondering what if in cases where two distinctively different cells which have same cell area, thickness, and volume can be differentiated? Can refractive index and cell structure be used in that case? I think it’s a key part that needs to be addressed in this paper to apply DHC across a vast range of cell types.
Reviewer 2 Report
The authors present a diagnostic tool (based on digital holographic microscopy) able to discriminate between breast cancer cells and white blood cells. Their results shown significant differences between morphological parameters, beating some limitations of other traditional methods.
The abstract, introduction, materials, methods and discussion are well-described. However, I think that a new revision of the results and conclusions could improve the quality of the work. On the one hand, the addition of some explanations (between lines 113-lines 137) could improve the understanding of the work. On the other hand, the addition of some references could make easier to compare the work presented vs the previous publications in this area.
Here I detail some minor changes that I think could improve the final version:
Line 30: Is the DHC a “new” powerful diagnostic tool?
Line 74: The breast cancer cell lines are not in the same order than in the table 1.
Line 101: Could you describe the cell incubator conditions (humity and temperature)?
Line 122: “EpCAM and CD45 expression on different cancer cell lines” Jurkat and THP-1 are blood cell lines or cancer cell lines also?
References: The detection of CTC are a hot-topic, is the bibliography updated?
Round 2
Reviewer 1 Report
The Author answered / updated all relevant questions asked. But there are still two things caught my attention which must be fixed before it can be approved for publishing
1. Figure 1 needs to be of high resolution. The copy of manuscript I downloaded from MDPI website has image which becomes pixelated upon zooming in. Also, Fig 1 should have appropriate labels on the figure for each part of the image.
2. I see discrepancy in reporting references. Such As
- Some reference do have DOI, some don't
- A lot references reported here has mismatch in capitalization for journal name, such as "Cancer epidemiology" in ref 1 and "Genome Medicine" in Ref 36.
So, the reference section must be carefully reviewed to fix mismatches. I am pretty sure the Editor will also ask to fix them before publishing. I will suggest to use commercial reference management software to prepare reference section, such as Endnote.
These two comments I made will be easy to fix and I believe the paper would be in better shape to be published.
